# Management of Low-Risk Thyroid Cancers: Is Active Surveillance a Valid Option? A Systematic Review of the Literature

**DOI:** 10.3390/jcm10163569

**Published:** 2021-08-13

**Authors:** Renato Patrone, Nunzio Velotti, Stefania Masone, Alessandra Conzo, Luigi Flagiello, Chiara Cacciatore, Marco Filardo, Vincenza Granata, Francesco Izzo, Domenico Testa, Stefano Avenia, Alessandro Sanguinetti, Andrea Polistena, Giovanni Conzo

**Affiliations:** 1PhD ICHT, 80131 Naples, Italy; 2Department of Advanced Biomedical Sciences, University of Naples “Federico II”, 80131 Naples, Italy; nunzio.velotti@gmail.com; 3Department of Clinical Medicine and Surgery, University of Naples “Federico II”, 80131 Naples, Italy; stefania.masone@unina.it; 4Department of Cardiothoracic Sciences, University of Campania “Luigi Vanvitelli”, Division of General and Oncologic Surgery, Via Pansini 5, 80131 Napoli, Italy; aleconzo@hotmail.it (A.C.); luigiflagiello.93@gmail.com (L.F.); cacciatore.chiara@virgilio.it (C.C.); giovanni.conzo@unicampania.it (G.C.); 5Division of Endocrine Surgery, Department of Oncological and Gastrointestinal Surgical Science, University of Study of Padova, Via Giustiniani 2, 35128 Padua, Italy; marcofinly43@gmail.com; 6Radiodiodiagnostic Unit, “Istituto Nazionale Tumori IRCCS Fondazione Pascale-IRCCS di Napoli”, 80131 Naples, Italy; v.granata@istitutotumori.na.it; 7Hepatobiliary Surgical Oncology Unit, “Istituto Nazionale Tumori IRCCS Fondazione Pascale-IRCCS di Napoli”, 80131 Naples, Italy; f.izzo@istitutotumori.na.it; 8Otolaryngology-Head and Neck Surgery Unit, Department of Cardiothoracic Sciences, University of Campania “Luigi Vanvitelli”, 80100 Naples, Italy; domenico.testa@unicampania.it; 9Scuola di Specializzazione in Chirurgia Generale, Università degli Studi di Perugia, 06123 Perugia, Italy; s.aveni@gmail.com; 101SC Chirurgia Generale e Specialità Chirurgiche, Azienda Ospedaliera S. Maria, 05100 Terni, Italy; a.sanguinetti@aospterni.it; 11UOC Chirurgia Generale e Laparoscopica, Dipartimento di Chirurgia Pietro Valdoni, Sapienza Università di Roma, Policlinico Umberto I, 00185 Roma, Italy; apolis74@yahoo.it

**Keywords:** thyroid cancer, active surveillance, thyroid surgery

## Abstract

Thyroid cancer is the most common endocrine malignancy, representing 2.9% of all new cancers in the United States. It has an excellent prognosis, with a five-year relative survival rate of 98.3%.Differentiated Thyroid Carcinomas (DTCs) are the most diagnosed thyroid tumors and are characterized by a slow growth rate and indolent course. For years, the only approach to treatment was thyroidectomy. Active surveillance (AS) has recently emerged as an alternative approach; it involves regular observation aimed at recognizing the minority of patients who will clinically progress and would likely benefit from rescue surgery. To better clarify the indications for active surveillance for low-risk thyroid cancers, we reviewed the current management of low-risk DTCs with a systematic search performed according to a PRISMA flowchart in electronic databases (PubMed, Web of Science, Scopus, and EMBASE) for studies published before May 2021. Fourteen publications were included for final analysis, with a total number of 4830 patients under AS. A total of 451/4830 (9.4%) patients experienced an increase in maximum diameter by >3 mm; 609/4830 (12.6%) patients underwent delayed surgery after AS; metastatic spread to cervical lymph nodes was present in 88/4213 (2.1%) patients; 4/3589 (0.1%) patients had metastatic disease outside of cervical lymph nodes. Finally, no subject had a documented mortality due to thyroid cancer during AS. Currently, the American Thyroid Association guidelines do not support AS as the first-line treatment in patients with PMC; however, they consider AS to be an effective alternative, particularly in patients with high surgical risk or poor life expectancy due to comorbid conditions. Thus, AS could be an alternative to immediate surgery for patients with very-low-risk tumors showing no cytologic evidence of aggressive disease, for high-risk surgical candidates, for those with concurrent comorbidities requiring urgent intervention, and for patients with a relatively short life expectancy.

## 1. Introduction

Thyroid cancer is the most common endocrine malignancy, representing 2.9% of all new cancers in the United States (US). It has an excellent prognosis, with a five-year relative survival rate of 98.3%, and has a higher frequency in females than males, with ratio of 3:1 [1]. Differentiated thyroid cancer (DTC) represents 90% of all thyroid malignancies and includes three main types: papillary thyroid cancer (PTC), the most common type, comprising 85% of all DTC; follicular thyroid cancer (FTC); and the rare subtype, Hürthle (oncocytic) cell thyroid cancer (2–5%) [2].

In the last ten years, the incidence of DTC has dramatically increased. This tendency is mainly as a result of the diffusion of imaging systems, the use of ultrasound-guided Fine Needle Aspirations Cytology (FNAC), and improvements in histological evaluations [3,4].

Actually, papillary microcarcinomas (PMCs) represents the most diagnosed thyroid tumors, with a 35% incidence of occult papillary thyroid microcarcinomas in autopsy studies [5]. The increased diagnosis of these malignancies, associated with a low risk of recurrence and death, has led to the need for redefining of the multimodal therapeutic approach to avoid potential overtreatments. With regard to treatment, historically, the only option was surgery. In the last few years, active surveillance (AS) has been established as an alternative approach; it is aimed at identifying patients who would likely benefit from rescue surgery [6].

Considering the data in the literature and the available evidence, we reviewed the current management of low-risk DTC, and PMCs in particular, to better clarify the indications for active surveillance for low-risk thyroid cancers.

## 2. Materials and Methods

According to the PRISMA flowchart (Preferred Reporting Items for Systematic reviews and Meta-Analyses), a systematic search was performed of electronic databases (PubMed, Web of Science, Scopus, and EMBASE). We used medical subject headings (MeSH) and free-text words, using the following search terms in all possible combinations: “differentiated thyroid cancer”, “micro papillary cancer”, “management”, and “active surveillance”. The last search was performed in May 2021. Attention was focused on the following primary outcomes: growth of the primary tumor, metastatic disease (lymph node or extra nodal), tumor recurrence after delayed thyroid surgery (DTS), and thyroid-cancer-related mortality. The secondary outcomes selected were decreased volume of primary tumor (>3 mm), overall mortality, and incidence of/indication for thyroidectomy.

The retrospective application of the surveillance criteria to patients surgically treated for thyroid nodules was not one of inclusion criteria for the studies reporting on AS of low-risk PTC; AS was limited to employment of surveillance strategies. Low-risk PTC was defined as T1a or T1b, N0, M0 disease. The search strategy was limited to articles written in the English language; moreover, papers on animal studies, review articles, editorials, and case series were excluded.

R.P. and G.C., two independent authors, analyzed all the papers, selected the suitable manuscripts, and performed the data extraction independently. All duplicate studies were removed. Two other authors (N.V. and S.M.) then checked the eligibility of the studies selected. Discrepancies were resolved by consensus.

## 3. Results

We identified a total of 2976 articles, of which 87 articles were selected for full text review. After full text review, 14 studies were included for the final analysis. The results are summarized in the PRISMA flowchart (Figure 1) [7,8,9,10,11,12,13,14,15,16,17,18,19,20].

The authors used the Newcastle-Ottawa Scale (NOS) to certify the quality of each included study. A maximum of nine stars was assigned to each study (Table 1).

A total number of 4830 patients under AS were included in this review.

All studies assessed tumor growth during AS; a total of 451/4830 (9.4%) patients experienced an increase in diameter of up to 3 mm. Conversely, in five studies, a decrease in tumor size >3 mm was assessed in 172/1324 (12.9%) patients during AS.

DTS after AS was performed in 609/4830 (12.6%) patients, as reported by all authors. Ten authors reported involvement of cervical lymph nodes during AS in 88/4213 (2.1%) patients, while in six studies, 4/3589 (0.1%) patients were reported to have extra-nodal metastatic disease. No study reported mortality due to thyroid cancer during AS.

The results are summarized in Table 2.

## 4. Discussion

DTC represents a heterogeneous pattern of different histological thyroid neoplasms [21]. With regard to histopathology, the most frequent variant is PTC, FTC represents 10–20% of cases, and medullary cancer is the rarest [22].

While the incidence of DTC continues to increase, mortality from thyroid cancer has declined over the last three decades. The prognosis for PMC is excellent, with a mortality rate of 0.3%; this includes patients with lymph node metastases and extra thyroidal extensions [23].

The mainstays of DTC therapy are surgical resection, radioiodine ablation, and thyroid-stimulating hormone (TSH) suppression; however, the benefits of active surveillance for selected cases have also been increasingly acknowledged in recent years [24,25]. Moreover, in 2016, encapsulated follicular variant papillary thyroid cancers (EFVPTC) were renamed non-invasive follicular thyroid neoplasms with papillary-like nuclear features (NIFTP) in an effort to avoid overtreatment arising from a diagnosis of cancer; the indolent nature of these neoplasms does not necessitate complete thyroidectomy or radioiodine ablation therapy [26].

The main risk factors for lymphatic recurrence are multifocality, infiltration of the thyroid capsule, positive margins, age, tumor size, and mutations of p53 or BRAF. The recurrence rate after surgery (thyroid lobectomy or total thyroidectomy) is 1–2% in all patients affected by unifocal PMC and less than 1% in patients without locoregional metastases. Conversely, sub-capsular location of the lesion, multifocal disease, and extrathyroidal microscopic extension are associated with a higher risk of recurrence and nodal metastases [27,28]. Many authors have shown that locoregional nodal metastases are also related to the male sex [29]. An age <45 years at diagnosis is considered as a risk factor for higher recurrence rate, as demonstrated in a recent meta-analysis [29,30].

Different prognostic scores have been proposed for DTC during the last ten years, taking into account the following elements:-biomolecular features (BRAF status) [31,32]; while TERT-promoter mutations can be found in less than 10% of PMC, BRAF seems to be the most reliable as a predictor of natural behavior of the PMC [31]. BRAF mutations can be found in 30–67% of PMC patients and are related to nodal metastases, extrathyroidal extension, and higher risk of recurrence [33];-histological features (fibrosis, distance between the lesion and the gland capsule, psammoma bodies) [23,32,33,34];-tumor pathologic factors, such as cervical lymph node metastases, tumor size, multifocality, and extra-thyroid extension [32,33,34];-patient factors such as male gender [34,35].

Based on these parameters, the scientific literature includes numerous studies in which patients with low-risk PTCM were safely and effectively managed with AS.

Furthermore, it has to be considered that thyroid surgery is not immune to complications; laryngeal nerve injury, hypoparathyroidism, and hypothyroidism are the most frequent adverse effects of total thyroidectomy, while post-surgical hematomas and surgical wound infections represent minor complications [36].

Currently, AS is not supported by ATA guidelines as the first-line treatment in patients with PMC, but is rather seen as an effective alternative, especially in high-risk surgical patients or in patients with more comorbid conditions. Molinaro and colleagues [37] confirmed the feasibility and safety of the AS approach for patients with very-low-risk tumors. High surgical risk, relatively short life expectancy, and cytologic biopsy with no aggressive disease were the characteristics for authors to identify patients eligible for AS.

In 2021, Lohia et al. [38] investigated patients >65 years with T1, N0, M0 PTC who received surgery and created a competing risk model to define patient groups with life expectancies of less than 10 and 15 years. They found, in their study of 3280 patients, that older patients with comorbidities have limited life expectancies but excellent Disease Specific Survival (DSS) from low-risk PTC and concluded that incorporating life expectancy into management criteria would likely promote less aggressive treatment such as AS.

A recent study by Tuttle et al. reported that AS was an appropriate strategy for patients with PTC less than 1.5 cm in diameter and isolated BRAF V600E mutations [39].

Oda et al. [15] compared surgically treated PMC patients with those who underwent simple surveillance. They showed an excellent outcome in both groups; however, logically, unfavourable events would be significantly higher in the surgical group. 

The results of our review of 4830 patients confirm the safety and efficacy of AS. Only 9.4% of recorded patients experienced an increase of tumor diameter, and 12.6% of patients underwent delayed surgery after AS. Cervical lymph node metastasis was present in 2.1% patients, and only 0.1% had extra-nodal metastasis.

Recently, Jeon et al. [40] proposed an interesting prospective study for papillary thyroid microcarcinomas based on KoMPASS (Korean Multicenter Prospective cohort study of Active Surveillance or Surgery). This study enrolled patients with PTMCs from 6 to 10 mm in diameter, with a confirmed cytopathological diagnosis and no metastasis or extra thyroid involvement. From the results of this survey, we will able to establish a protocol for the clinical framework of patients eligible for AS.

As previously stated, it is of paramount importance to consider tumor characteristics, family history, age, and other risk factors for the appropriate selection of patients for AS; moreover, it is necessary to evaluate individual patient’s attitudes, including anxiety and reliability, as an effective AS approach is based on patients’ compliance with surveillance protocols and follow-up.

The possibility of disease progression must always be considered: tumor growth in a short time period or the onset of lymph node metastases must be diagnosed promptly in order to switch patients to surgery without delay.

In our opinion, in the future, we will be able to use a prognostic score in order to determine those patients who can be treated using AS for PTC; this will allow for a concrete therapeutic option to minimize surgical intervention risk for non-aggressive tumors.

Hospitals and regional or national health organizations are increasingly interested in cost-effective patient management. In this systematic review, unfortunately, we are not able to provide a cost analysis of active surveillance versus standard therapy. In addition, we did not find any studies during our literature review that focused on this topic.

In our opinion, it would be interesting to perform a cost analysis of physician review and molecular testing compared with the standard therapy. This will be a focus of our research in the near future.

Finally, the challenges presented by the COVID-19 pandemic further highlight the benefits of AS in minimizing surgeries in a context undermined by the possibility of contagion [41,42,43].

The limitations of the present study include its design: because no statistical analysis was carried out on the extracted data, no definitive conclusions can be reached. Further research, with a meta-analytic design, is needed to evaluate the weighted incidence of tumor growth and local/distant metastasis during AS in the included studies, thus avoiding possible bias related to patient demographics and tumor characteristics.

## 5. Conclusions

Active surveillance should be proposed for low-risk PMC only after clinical trials proving the validity of this approach. For high-risk PMC, (lymph node or distant metastasis, extra thyroid extension, closeness to recurrent laryngeal nerve or trachea, high-grade cytology, or growth during observation), a surgical approach (lobectomy with or without paratracheal dissection) is necessary. Better knowledge of papillary cancer natural history and biological behavior might be useful in the design of multimodal management.

## Figures and Tables

**Figure 1 jcm-10-03569-f001:**
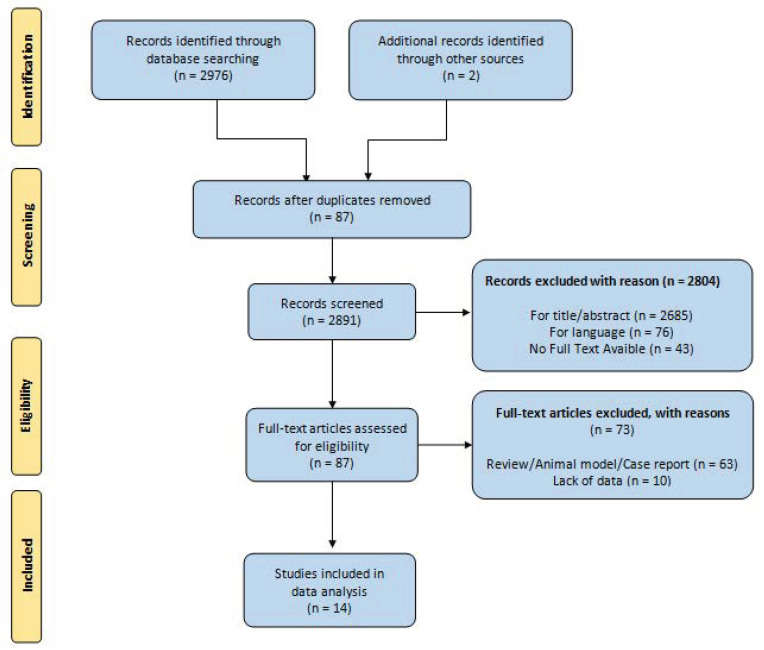
PRISMA flowchart.

**Table 1 jcm-10-03569-t001:** Newcastle-Ottawa Scale (NOS).

References	Selection				Comparability	Outcome Assessment		
	1	2	3	4	1	1	2	3
Gorshtein, 2021	*	*	*	*	**	*	*	
Rozenbaum, 2021	*		*	*	*	*	*	
Hu, 2021	*	*	*	*	**	*	*	*
Rosario, 2019	*	*	*	*	**	*	*	*
Oh, 2019	*	*	*	*	**	*	*	
Kim, 2018	*		*	*	*	*	*	*
Fukuoka, 2016	*	*	*	*	*	*		
Ito, 2014	*	*	*	*	**	*	*	*
Tuttle, 2017	*	*	*	*	**	*		
Oh, 2018	*		*	*	**	*	*	
Kwon, 2017	*		*	*	**	*	*	
Sugitani, 2010	*	*	*	*	*	*	*	
Sanabria, 2018	*	*	*	*	*	*	*	*
Oda, 2016	*		*	*	*	*	*	

Identify high quality choices with a “star”: *. A maximum of one “star” for each item within the “Selection” and “Outcome Assessment” categories; maximum of two “stars” for “Comparability”.

**Table 2 jcm-10-03569-t002:** Characteristics of included studies.

First Author, Year	Patients, *n*	Age, Years	Female Sex, %	Tumor Growth, *n*	Delayed Surgery, *n*	Cervical Lymph Node Metastases, *n*	Localization of Metastasis	Distant Metastases, *n*	Death
Gorshtein, 2021	189	60.1 ± 13.1	89.2%	23	19	-	-	-	-
Rozenbaum, 2021	80	-	-	24	16	-	-	-	-
Hu, 2021	212	-	-	39	101	17	Central	2	-
Rosario, 2019	12	53 (median)	81.2%	1	2	1	-	-	-
Oh, 2019	273	51.1 (median)	75.8%	77	52	17	Central	-	-
Kim, 2018	126	51 ± 7	78.7%	25	18	1	-	-	-
Fukuoka, 2016	384	54 ± 11.9	86.1%	29	15	12	Central	-	-
Ito, 2014	1235	-	90%	58	191	19	-	-	-
Tuttle, 2017	291	51 ± 23	75.3%	11	5	-	-	-	-
Oh, 2018	370	51.1 ± 11.7	76.8%	86	58	5	-	-	-
Kwon, 2017	192	51.3 (median)	76 %	27	24	7	Central	2	-
Sugitani, 2010	230	56 (median)	50%	22	9	3	Central	-	-
Sanabria, 2018	57	51.9 (median)	84 %	2	5	-	-	-	-
Oda, 2016	1179	57 (median)	87 %	27	94	6	-	-	-

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
