# Peer review of "Management of Low-Risk Thyroid Cancers: Is Active Surveillance a Valid Option? A Systematic Review of the Literature"

_jcm, 2021, doi:10.3390/jcm10163569_

Round 1
Reviewer 1 Report
The Authors present a systematic review on active surveillance for low-risk thyroid cancers. Fourteen studies were included in the review for a total of 4830 patients.
The study is well conducted and the criteria of the Preferred Reporting Items for Systematic reviews and Meta-Analyzes (PRISMA) were met. The main scientific databases for the sector were consulted. A careful selection was made.
The conclusions are the right consequence of the analysis of the data and in particular the prudence in the final considerations should be appreciated. Active surveillance must be relegate only to those cases without suspicious extra-thyroidal extension or metastasis or multiple thyroid nodules or other thyroid diseases which require a total thyroidectomy.
In any case I agree with the statement that “…active surveillance can be an alternative to immediate surgery in patients with very low-risk tumors such showing no cytologic evidence of aggressive disease, in high-risk surgical candidates, those with concurrent comorbidities requiring urgent intervention, or patients with a relatively short life expectancy.”
Finally it is rightly emphasized that the active surveillance approach must be based on patients' compliance to surveillance protocols and follow-up.
Typographical corrections
Title page - “…systematic…” instead of “…sistematic…”
Page 4 – Line 107 - “…were recorded…” instead of “…was recorded…”
Page 5 – Line 168 – “…bilateral vocal cord injury…” instead of “…bilateral cord vocal injury…”
Author Response
Review 1:
First of all, thank you for your appreciate work and your consideration for our manuscript.
We are so proud about your valuation and we hope our work may be a motivation to improve the Active surveillance program.
We had checked our manuscript as your review:
- we correct the sentence in the title page
- we complitely change the sentence in page 4
- we change the “…bilateral vocal cord injury…” instead of “…bilateral cord vocal injury…” in page 5
We are sure that your appreciate advise have made more readeble our paper.
Thank you.
Reviewer 2 Report
In this paper, the group perform a systematic review of studies on active surveillance of small, low risk papillary thyroid cancers. This is an interesting topic and I am delighted to see this academic group addressing it.
Overall the paper is nicely written although there are a few typographic errors (eg 'sistematic' in the title) and the results are somewhat brief.
Results section:
I would like to read some more granular detail in the results extracted from each paper if possible eg 1 - within the individual studies what were their inclusion criteria (this is important to get a sense of any possible bias), 2 - demographic data from the patients in each study included, 3 - were the lymph node mets in the 2.1% clinically relevant?
Discussion section:
The decision to submit a patient to AS vs surgery is about balancing the risks vs benefits of each. The authors describe the natural history of these published patients undergoing AS however, any clinical risk of disease progression is unclear.
Additionally the type and risks of surgery seem overstated in this instance as the alternative to AS for papillary microcarcinoma would usually be hemithyroidectomy not total, with RAI and TSH supression.
The issue of bias in studies of AS needs to be specifically addressed: do these patient's demographics differ from those in surgical studies or the expected demographics of patients with PTMC?
Health economics/cost effectiveness - this is an important issue which needs to be addressed: how do the costs of serial ultrasound, physician review and molecular testing compare with a hemithyroidectomy?
I would be happy to review an updated version of this paper.
Author Response
Reviewer 2:
First of all thank you for your attention and your appreciate review of our manuscript.
We are so proud of your word about our paper.
I will answer point by point to your request and review.
RESULT SECTION:
I would like to read some more granular detail in the results extracted from each paper if possible eg 1 - within the individual studies what were their inclusion criteria (this is important to get a sense of any possible bias), 2 - demographic data from the patients in each study included, 3 - were the lymph node mets in the 2.1% clinically relevant?
Thank you for suggestion; granular data such as demographic characteristics and localization of neck metastasis, when present in included study, have been added to Table 2.
About inclusion criteria, they are homogeneous in all included studies as stated in Methods section: “Inclusion criteria regarded all studies reporting on AS of low-risk PTC with AS limited to employment of surveillance strategies and not the retrospective application of surveillance criteria to patients treated surgically for thyroid nodules. Low-risk PTC was defined as T1a or T1b, N0, M0 disease”
DISCUSSION SECTION:
The decision to submit a patient to AS vs surgery is about balancing the risks vs benefits of each. The authors describe the natural history of these published patients undergoing AS however, any clinical risk of disease progression is unclear.
Additionally, the type and risks of surgery seem overstated in this instance as the alternative to AS for papillary microcarcinoma would usually be hemithyroidectomy not total, with RAI and TSH supression.
Thank you for suggestion. In discussion section risk of progression has been now clearly addressed (see page 5 Line 31)
The issue of bias in studies of AS needs to be specifically addressed: do these patient's demographics differ from those in surgical studies or the expected demographics of patients with PTMC?
Thank you for suggestion; because of no statistical analysis has been carried out on demographics characteristics of included patients, no final conclusions on this aspect could be designed. This possible bias has been addressed in the limitations of the study present at the end of the discussion section.
Health economics/cost effectiveness - this is an important issue which needs to be addressed: how do the costs of serial ultrasound, physician review and molecular testing compare with a hemithyroidectomy?
Thank you for interesting reflection about health economics/cost effectiveness, we think that a complete analysis must valuate that implications. Unfortunately we have no data about and no study in Literature focused on this topics. We added informations about this topics at the page 5 from line 36.
We upload a new revised version of the manuscript. We hope we have done a good job.
Kind regards